# Scent of a Woman—Or Man: Odors Influence Person Knowledge

**DOI:** 10.3390/brainsci11070955

**Published:** 2021-07-20

**Authors:** Nicole L. Hovis, Paul R. Sheehe, Theresa L. White

**Affiliations:** 1Department of Psychology, Le Moyne College, Syracuse, NY 13214, USA; HovisNL@Lemoyne.edu; 2Human Services Department, Capella University, Minneapolis, MN 55402, USA; 3Neuroscience and Physiology, SUNY Upstate Medical University, Syracuse, NY 13210, USA

**Keywords:** odorant, olfaction, impression formation, masculinity, femininity, gender

## Abstract

First impressions of social traits are regularly, rapidly, and readily determined from limited information about another individual. Relatively little is known about the way that olfactory information, particularly from scents that are not body odors, alters a first impression. Can the attributes of an odorant be conferred onto a person associated with that scent? To explore this, 101 participants were asked to form an impression of a hypothetical person based on the following stimuli: A gender-neutral silhouette, a list of six personal characteristics, and one of five odorants. Participants then rated the likelihood that the hypothetical person possessed each of 51 personality traits that were determined a priori as falling into six attribute categories. Participants also directly rated all odorants for the six categories and intensity. A T-test showed that ratings of the hypothetical person were less disparate from the odor that was presented during impression formation than from other odors. ANOVA revealed that the effects were heterogeneous, with odorants varying in their effectiveness in associating the hypothetical person with categories. The present data suggest that a hypothetical person can be imbued with the specific attributes of an odor and that some odors are better at contributing to impressions than others.

## 1. Introduction

Friend or foe? Mate or kin? Man or woman? In order to make such decisions in initial social encounters, people rapidly form first impressions of others by processing a limited amount of information coming from their senses [1,2,3]. Humans are sensitive to social cues that have implications for survival and reproductive fitness [4]; thus, the first appraisals that people make of novel individuals likely reflect those critical imperatives. Each of the senses contributes what it can; vision, audition, and even the sense of smell may provide incoming information about the new person. The available physical information is quickly processed by neural substrates, e.g., [5,6], and the incoming information is contextualized in the light of past experiences in other brain areas, such as the dorsomedial prefrontal cortex (dmPFC) [7,8], along with social aspects of the novel individual that can be discerned such as gender, emotional expression, or perceived dominance [9,10,11,12]. Neural imaging shows that the dmPFC also responds to body odors, suggesting that odors also give information that contributes to processing a social scene [13,14] and the formation of a first impression.

The way that people smell (body odor) likely serves an adaptive function [15] and can be a valuable source of social information [16,17,18]. People emit odors that reflect personal characteristics from the axillary areas, breath, eyes (through tears), and hands [16]. These odors are unique to an individual [15,19] and can be used to infer that person’s identity [14,20]; for example, a mother can recognize her child by its scent, e.g., [21]. Some enduring physical traits, such as biological sex, are reflected in a person’s body odor [22]; smells arising from both the breath [23] and the underarms [24,25,26] can be correctly categorized as originating from a man or a woman, though this is not consistently observed [27]. When people are able to identify another individual’s sex using the sense of smell, the odors judged as coming from male body odor donors are generally rated as less pleasant [23] and more intense [25], suggesting that these two attributes might underlie the strategy for categorization.

In addition to enduring traits, temporary circumstances such as emotional state [22] or diet [28,29] can also be reflected in a person’s body odor. Transitory states, such as illness, fertility, or sexual interest [30,31,32], can be detected through body odor. These transitory states can affect the way that individuals are perceived by others; for example, the scent of a sick person is judged to be coming from someone who is less likable [32].

Although sweat-based odor is not fully blocked by the use of fragrances and thus is considered “honest” as a reproductive cue [33], much of Western society attempts to disguise their natural scent [20,34] and to increase their own attractiveness [35,36] through the use of antiperspirants and perfumes. The dimension of gender (as a social construct) seems to be one of the most important dimensions of an odor for perfumers [37], and although the distinction seems to be more of an overlapping continuum than a simple distinction between masculine and feminine [38], it is a relevant one for the development of new fragrances.

Social judgments based on body odorants are likely also informed by fragrance components, as well as underlying genetic, health, and dietary information [33]. Day-to-day body odors are influenced by dietary choices [28,29,39], leading to variability. For example, odors from monozygotic twins on different diets can be discriminated against but not if they are on the same diet [40,41]. In many studies involving body odor, food is controlled in order to avoid its potentially confounding or overwhelming effects; participants are asked to avoid strong-smelling foods, such as garlic or blue cheese [28,42]. One of the questions asked by the present study is whether food odors could act as a proxy for body odorants in triggering social judgments about gender.

The environmental cues to gender that signal masculine or feminine stereotypical constructs are quite diverse: Vocal quality [43,44], facial structure [45], facial coloring [46], handwriting [47], hairstyle [48], clothing [49,50], and the color of clothing or other objects associated with individuals [51]. The concept of masculinity or femininity that is marketed with a perfume seems able to trigger cross-modal odor-color gender stereotypes as well; a series of studies have demonstrated that when people smell a fine fragrance, a specific color comes to mind that differs based on whether the perfume is marketed to men (blues or greens) or women (pinks or yellow) [52]. As color can be used as a cue to gender, e.g., [53], this is an indication that it is possible that odorants, even those that do not solely originate from the body, may be used as a similar associative cue.

Very few studies directly assess whether the environmental odorants that are often associated with people, such as fine fragrances or food odors, might assist in the formation of first impressions about attributes such as gender, and the evidence that the research provides is somewhat contradictory. At least two studies indicate that fine fragrance can indeed alter the perception of masculinity or femininity [54,55]. One study reported that a hypothetical person who was wearing a floral fragrance was rated as having fewer male characteristics [54], while the other showed that the presence of a typically masculine perfume increased the employability of a potential job applicant, presumably through triggering a concept of maleness and power [55]. Another study suggested that food odorants (in addition to perfumes) may indicate the masculinity or femininity of a hypothetical person since it showed that the smell of a lemon was associated with femininity and cleanliness and that the scent of onion was associated with masculinity and a lack of hygiene [56]. The concept of food odors conveying gender information seems more controversial, however, as people who could discriminate between the body odors of men who had eaten meat (or not) nevertheless rated the masculinity of the odor samples similarly [28,57], suggesting a stability for this trait that could be unaffected by dietary components. In essence, although the few studies that have directly presented environmentally-based odorants as information relevant to the formation of an impression of a person suggest that the olfactory information may influence judgments, the case is not particularly strong or clear. In addition, none of the studies that directly assessed the effects of environmentally originating odorants on gender impression formation have shown directly that that people use this type of olfactory information in assessing a novel individual or whether the attributes associated with the odor itself were imbued upon the impression of the new person. To address these remaining questions, the present study directly asks whether people take environmental olfactory information into account when forming an impression of another person—particularly when drawing a conclusion regarding gender. In the present experiment, participants were asked to form an impression of the hypothetical person based on a brief presentation of visual, verbal, and olfactory stimuli; the visual and verbal information did not differ, but each participant received only one of five possible odorants to use in forming the impression. After participants reported that they had formed an impression of the new person, they were presented with a series of more than 50 traits that had been pre-selected to fall into attribute categories (Clean/Unclean, Pleasant/Unpleasant, Feminine, Masculine) and asked to rate each trait as to how likely the hypothetical new person was to possess it. Following this assessment, participants directly rated all five of the olfactory stimuli on the same attribute categories that had been indirectly assessed (via specific traits) in relation to the hypothetical person. The principal hypothesis of the present study was that changing an odor associated with a hypothetical person would change the perception of the traits associated with that person, including their gender, in such a way that if an odorant was perceived as having certain attributes, then when it was paired with a hypothetical person, that person would be imbued with those attributes. Evidence in support of this hypothesis would be that the ratings of the traits of that person would be consistent with the ratings of the odor presented at the time of impression formation rather than uniformly across odorants. Thus, it was hypothesized that, on average, the descriptor ratings of the hypothetical person would be closer to the attribute ratings of the odorant associated with that person at the time of impression formation (*same odor*) than to the ratings of the other odorants (*other odors*).

## 2. Materials and Methods

### 2.1. Participants

This study involved 109 (17 Males, 91 Females) undergraduate students (*M* age = 20.26, *SD* = 4.48) at Le Moyne College who took part in the exchange for extra credit in a psychology course. Eight participants (all females) were excluded because they did not follow directions. Thus, 101 people fully participated in the experiment. All participants were interviewed to ensure that they felt that they had a normal sense of smell and did not suffer from multiple chemical sensitivity prior to participation. Participants were treated in accordance with the Le Moyne IRB approved protocol, and each signed a written informed consent document. This study complies with the Declaration of Helsinki for Medical Research involving Human Subjects.

### 2.2. Materials

In this study, the five olfactory stimuli (all provided by International Flavors and Fragrances) of similar intensities were presented inside of small plastic containers via odor-infused tablets that were covered by cotton that acted as a filter and a visual shield. Two of the olfactory stimuli were food odors (lemon and onion), two were fine fragrances (amber and lily), and one was a no odor control. The two fine fragrances were recommended by International Flavors and Fragrances research scientists as being highly representative of masculine (amber) and feminine (lily) fragrances (K. Rankin, personal communication, 22 April 2009).

Two laminated sheets comprised the visual stimuli: One depicted a silhouette (see Figure 1) of a person [58], while the other listed six personality characteristics: intelligent, skillful, industrious, determined, practical, cautious [1].

The program created in E-Prime software (Psychology Software Tools, Inc., Sharpsburg, PA, USA) for this study was presented to each participant on a Dell Optiplex Gx620 computer. The program presented each of the personality traits listed in Table 1 on a screen singly, in a random order, and waited for a rating response from the participant before moving along to the next trait. Rating responses were gathered using a standard computer keyboard.

### 2.3. Procedure

Each participant was assigned to one of five olfactory stimulus groups using a randomized block design: Odor of lemon, odor of onion, odor of amber, odor of lily, and a no odor control. The olfactory stimulus groups varied in size; there were 19 participants in the group of people who received the lemon odor (see below), 21 in the onion group, 21 in the amber group, 19 in the lily group, and 21 in the control group. Other than the olfactory stimulus, the groups did not differ in procedure. Each participant was taken to a room and seated in front of a computer, then presented with three types of contextual stimuli: The laminated picture of a gender-neutral silhouette [58] depicting a hypothetical person (Figure 1), the laminated list of the six personality characteristics belonging to the hypothetical person [1], and a plastic container containing an olfactory stimulus (varying by group). The participant was asked to form an impression of the hypothetical person based on the silhouette, the list of characteristics, and the odor. At this point, participants could investigate the stimuli for as long as they liked, though no one exceeded 4 min of inspection. It was hoped that the olfactory and visual stimuli (the silhouette and the list of six personality characteristics) would form a context for consideration of the hypothetical person.

Immediately after the participant indicated to the researcher that a strong impression of the hypothetical person had been formed, the contextual stimuli were removed from the testing room. The participant then received instructions (via computer) to indicate how well each of the personality traits that appeared on the screen would describe the hypothetical person on a series of 9-point scales (1 = *not like the person,* 9 = *very like the person*). In short, the participant assessed their impression of the person on each of the 51 personality traits [1,54] listed in Table 1, which had been determined a priori by the researchers to fall into six different categories (or attributes): clean, unclean, pleasant, unpleasant, masculine, or feminine (see Table 1). Each trait was presented one at a time, centered on the computer screen in front of the participant. After the participant rated the trait, a new trait would appear in its place until the participant had rated all 51 of them. These personality traits were presented in a different random order for each participant.

Regardless of which odorant they experienced during impression formation, after completing the assessment of the hypothetical person, each participant evaluated all four olfactory stimuli, one at a time, in a random order on a number of psychophysical measures. The participants rated each odorant on variations of the gLMS (general labeled magnitude scale) for intensity [59] and pleasantness [60]. Each odorant was also rated for femininity, masculinity, and cleanliness using visual analog scales (2.25 cm) that were anchored at one end by “*Not at all*” and at the other end by “*Extremely*”. Participants were orally debriefed at the end of the study and given a written debrief to take with them.

## 3. Results

### 3.1. Standardization of Ratings

The ratings of the personality attributes associated with the hypothetical person were transformed to fit into four categories rather than six for a more direct comparison with the ratings of the olfactory stimuli. Though masculinity and femininity were kept as separate categories [38,61], the remaining four categories (pleasant, unpleasant, clean, unclean) were reduced to two categories (pleasantness, cleanliness) that were reflected by 9-point scales by multiplying each descriptor rating of the hypothetical person from the unclean (or unpleasant) category by −1 and then applying the equation of y_1_ = (y_2_ × 8/18) +5, where y_2_ is the net total of the positive and negative ratings, and y_1_ is the transformed data point. The resultant four categories of personality attributes associated with the hypothetical person were then further transformed to fit a 0 to 100-point scale. Thus, each individual attribute rating of the hypothetical person was transformed using the equation y_1_ = (y_2_ − 1) × (100/8), where y_2_ again represented the original data point and y_1_ represented the transformed data point.

Because the psychometric scales used to rate the odorant qualities differed from each other, participant ratings of the odorants were also standardized to fit a 0 to 100 scale. This standardization made the odorant ratings directly comparable to ratings of the hypothetical person. Three different equations were employed in order to properly transform each individual odor rating to a 100-point scale, each of which included y_2_ as the original data point, and y_1_ as the transformed data point. For the scale of intensity, each odor rating was transformed using the equation y_1_ = y_2_ × (100/95). For the scale of pleasantness, the equation y_1_ = y_2_ × (100/200) + 50 was utilized. For the ratings for masculinity, femininity, and cleanliness, the equation y_1_ = y_2_ × (100/2.78) was used.

Thus, at the end of these transformations, the data from the ratings associated with both the hypothetical person and the odorants reflected four categories of attributes (pleasantness, masculinity, femininity, and cleanliness) that were represented on a 0 to 100-point scale. In this way, the relationships between the ratings of the hypothetical person and those of the odorants were expressed in comparable scales.

### 3.2. Stimulus Group versus Control Ratings

An initial question from this experiment addressed whether the mere presence of an odorant produced a consistent change in the ratings of the traits that comprised the four attributes (pleasantness, masculinity, femininity, and cleanliness) associated with the impression of the hypothetical person. The trait means from each attribute were averaged over the four odor groups (Lemon, Onion, Lily, Amber) to produce a grand mean for each attribute. These grand means were then tested in comparison to the control (no-odorant) group means by Hotelling’s multivariate T-square test at the 5% level of alpha. The sensitivity of the test was optimized by using error variances with the effects of individual stimuli removed. The difference between the combined odor and the control group was not found to be statistically significant, *F*(4,96) = 0.49; *p* = 0.75, by Hotelling’s T^2^ criterion, indicating no general effect of the mere presence of an odorant.

### 3.3. Descriptive Measures

The mean and standard deviations of ratings for the attributes (pleasantness, masculinity, femininity, and cleanliness) as they pertain to both the hypothetical person and to the direct ratings of the four odorants are presented in Table 2 for each stimulus (odorant impression) group. The directly rated odorant that corresponds to the odorant presented at the time of impression formation (i.e., *same odor*) is shown in bold print for each group since this relationship forms the basis of the principal analysis.

The disparity between any two attribute rating profiles is also summarized in Table 2. This measure describes the disparity between the hypothetical person and each rated odorant and was calculated as the root mean square (RMS) deviation between the paired elements of the profiles. A close examination of Table 2 shows that the RMS for the odorant that corresponds with the smell presented during impression formation (i.e., *same*) is generally lower than for the other odorants. Further, note that, in all four odorant groups, the mean masculinity rating of the hypothetical person exceeds the mean masculinity rating of every odorant. This is true only for the masculinity attribute.

### 3.4. Analytical Tests

Disparities were calculated in two different ways for analysis. The *person-vs.-same* odorant disparity refers to the disparity between a person profile and the profile for the same odorant as the stimulus. The *person-vs.-other* odorant disparity indicates the average of the disparities between the person profile and the profiles of the three odorants other than the stimulus. Of particular interest was the difference between the *person-vs.-same* disparity and the *person-vs.-other* disparity, which was labeled for convenience as the *same-vs.-other* disparity.

The principal hypothesis postulated that the odor associated with a hypothetical person could change the perception of their personality traits, including their gender, in a way that was consistent with the odorant. The principal analysis tested whether the grand mean of the *same-vs.-other* disparities across all four odorant groups (control group excluded) was significantly less than 0 at the 5% one-tailed level of alpha. For validity and maximal power, the *t*-test used the error variance obtained in a one-way ANOVA across the four stimulus groups. A mean value that was significantly less than zero in the *t*-test would reflect, on average, a pattern of ratings of the hypothetical person more similar to that of the *same* odorant (presented at the time of impression formation) than to the patterns of ratings of the *other* odorants.

The principal question for this study was whether an odor associated with a hypothetical person would cause an impression of that person that was more similar to that odor (*person-vs.-same*) than to other odors (*person-vs.-other*) in terms of pleasantness, masculinity, femininity, and cleanliness. This was found to be the case, as the observed grand mean (*M* = −3.06, *SE* = 1.06) was significantly lower than 0, *t*(76) = −2.90; *p* = 0.002. This shows that, on average, participants rated the four attributes of the hypothetical person to be more similar to those of the *same* odor (lower disparity) than to the *other* odorants.

The principal analysis also included a 3 degrees of freedom *F*-test of the heterogeneity of the four stimulus group effects at the 5% level of alpha. The one-way ANOVA revealed substantial heterogeneity between the stimulus groups, *F*(3,76) = 3.66, *p* = 0.016, indicating that the *same-vs.-other* disparities, as shown in Table 3, varied significantly across stimulus groups. The grand mean, as well as the means for the Onion and Amber groups, were all significantly less than 0 at the one-tailed 0.01 level, while the means for the Lemon and Lily groups did not differ significantly from zero. Follow-up *t*-tests uncorrected for multiplicity indicated significant differences in two comparisons: between the lemon and onion stimulus groups, *t*(76) = 3.61, *p* = 0.0005, and between the lemon and amber stimulus groups *t*(76) = 2.53, *p* = 0.013.

### 3.5. Interpretive and Exploratory Analyses

The statistical techniques used in the analyses to be described here include linear regression, correlation coefficients, *t*-tests, and ANCOVAs. To preserve the essential meaning of the interpretive and suggestive findings, the variety of nominal significance tests and *p*-values were uncorrected for test multiplicity.

#### 3.5.1. Intensity Analysis

Because the intensity rating of an odorant has been reported to be related to some undefined function of the four attribute ratings, e.g., [25,56], it was of interest to see how much intensity ratings were involved in the results of the principal analysis. The grand mean intensity (and standard deviation) varied across odorants as follows: Onion, *M* = 38.10 (*SD* = 25.64); Lemon, *M* = 59.77 (*SD* = 22.39); Amber, *M* = 37.85 (*SD* = 24.66); Lily, *M* = 28.07 (*SD* = 22.49). To evaluate the level of intensity involved in the principal analysis, an ANCOVA of the *same-vs.-other* disparities was performed, with the stimulus groups as the factor and with the intensity ratings of all four odorants as the covariates. This analysis shows the means of the *same-vs.-other* disparities in the principal analysis in terms of two component values: The means estimated in the ANCOVA of stimulus group effects and the joint effects of the four stimulus intensity ratings. The ANCOVA estimates of *same-vs.-other* disparity means in the four stimulus groups are shown in Table 3. The differences between these means and those in the principal analysis, also shown in Table 3, represent the effects of the intensity covariates. By inspection, the estimated means show somewhat greater heterogeneity than the principal means, and the intensity effects are mild and variable. Note that since the estimated means were evaluated conventionally at the mean levels of the four intensity covariates, the grand means of *same-vs.-other* disparities for all four groups in the two analyses are necessarily identical, so no overall intensity effect was possible.

#### 3.5.2. Analysis of Relationships between the Attributes

The ratings of the four attributes (pleasantness, cleanliness, masculinity, and femininity) assigned by the participants (four stimulus groups as well as the control group) were analyzed for their relationships to each other in terms of correlation coefficients. The six possible Pearson’s correlation coefficients were calculated for the attribute ratings of the hypothetical person, and six were also calculated for the average ratings of the odorants. These correlation coefficients may be seen in Table 4. The correlation patterns were examined informally, and as can be seen in Table 4, many of the attributes were strongly related to each other, with only Masculinity and Pleasantness failing to show an association both in the person ratings and the odorant ratings.

The pattern of attribute correlations for persons was informally compared to the pattern for odorants. A visual inspection of Table 4 reveals that although the two different types of ratings generally (with a few exceptions) indicated relationships between the attributes that were similar in direction, they varied considerably in strength. The correlations derived from the two types of ratings differ most in three of the attribute relationships: Masculinity-Femininity, Masculinity-Cleanliness, and Femininity-Cleanliness. The most notable difference is the significant negative correlation between Masculinity and Cleanliness when rating odorants that were reversed to a significant positive correlation when rating the hypothetical person.

#### 3.5.3. Analysis of Odorant Influence

Since all four attributes contributed to each profile in the principal analysis, a natural question would be to ask which of the attributes was most imbued by the odorants in the ratings of the hypothetical person. To answer the question “Which attributes of the odorants had the largest influence on the results of the principal analyses?” the disparities between the participants’ individual attribute ratings for persons and odorants were used. (The disparity between individual attribute ratings reduces to the absolute value of the person-odorant difference.) Taking cleanliness as an example of an attribute, the *same-vs.-other* difference between cleanliness disparities was calculated for each participant strictly analogously to the *same-vs.-other* profile disparities described in the principal analyses. Finally, the correlation coefficient between the cleanliness and profile values was squared and multiplied by 100 to obtain the percent of variance in the profile values that was accounted for by cleanliness. For each attribute, the *same-vs.-other* difference in disparities was correlated to those of the *same-vs.-other* profile disparities so as to determine the relative amount of influence that each had on the principal analysis. The amount of influence of each attribute was as follows: Cleanliness, 0.36%; Pleasantness, 4.83%; Masculinity, 49.98%; Femininity, 12.28%. Note that the total of the four attributes does not equal 100%, but rather 67.44%, indicating that the inter-relationship of the four attributes may also have an important influence.

## 4. Discussion

The present study was conducted in an effort to further understand the role of odorants as social cues that are relevant to survival and reproduction by investigating whether odorants associated with the human body (but not produced by it) may nevertheless signal personal information, such as gender. Perhaps more importantly, this study also sought to examine whether the attributes associated with an odorant could be imbued upon a person who was associated with it in the course of a first impression. The results indicate that odorants are an important source of social information about another person; changing the odor associated with a hypothetical person changed the perception of the traits associated with that person, including their perceived gender. Further, people were assumed to have the attributes of odorants that were associated with them, as the descriptor ratings of the hypothetical person were closer to the attribute ratings of the odorant associated with them at the time of impression formation than to the ratings of the other odorants.

The present results support the idea that people rely on all the available information, both verbal and nonverbal, to make rapid judgments about other people [62,63]. When forming a first impression of someone, people have the opportunity to see, hear, and smell that individual. The ecological approach to impression formation [62] places emphasis on the role that information arising from direct interactions with an individual plays in determining how we think about them. When information arising from that individual is restricted, as in the present experimental situation, it is clear from the present data that odorants are considered when drawing conclusions about a person. This confirms previous research showing that activating the concept of an odorant verbally (such as vanilla) can alter the impression of another person [64]. The present data shows that the contribution of the olfactory information is not general, such as “this person stinks”; rather, the results of the principal analysis showed that the specific odor involved is important, with some odors being more successfully associated with a person than others. In particular, the amber perfume and the onion odorant were both apt to affect the impression of the hypothetical person, while the lily perfume and the lemon odorant were less likely to do so. This finding suggests that the category (food or fine fragrance) of the odorant’s common usage is not a relevant factor in determining whether or not the odorant will affect a first impression. Given the identity of the effective odorants, the level of masculinity associated with an odorant may be important to impression formation.

The present data shows that an odorant can imbue the hypothetical person with attributes similar to itself, even when differences in intensity levels are taken into consideration. This finding supports the idea that odors associated with a person are perceived to be an extension of that person’s body [24]; the present study extends this idea to odorants that are not naturally produced by people. In other words, even though the odorants in this study are not body odors, the fact that the attributes associated with the odorants became associated with the hypothetical person suggests that the participants treated the smells as if they were body odorants; they acted as though the attributes of the odorants belonged to the hypothetical person. The particular attributes associated with the odorants likely arose from past associations with the odor but may have been based on the similarity of olfactory qualities to stereotypical attributes of people (i.e., if the odorant is “delicate”, it might be associated with femininity, as this trait is stereotypically associated with females).

Although the construct of gender is increasingly recognized as complex and non-binary in American society, the categories of masculine and feminine are still organizing principles in social life, and people learn about cues that signal the distinction between masculine and feminine within a culture [50], beginning in childhood, e.g., [65]. The result of the present experiment suggests that information from odorants, even those that do not arise from the human body, can be one type of gender cue. The attribute of the odor that was most strongly related to the results of the principal analysis was masculinity, suggesting that this attribute could be more strongly imbued upon people by odorants. Thus, when the odorant was perceived as extremely masculine, the person associated with the odorant was considered to have a masculine gender identity. This finding is in keeping with previous work that showed that perfumes marketed as “masculine” increased the perception of a potential job applicant as a suitable candidate for a managerial position [55], a position of power. Within American culture, the attribute of masculinity is related to the concept of dominance and power, e.g., [55,66]. It is important that those in a subordinate position maintain awareness of those who are dominant so as to better understand the person in power and gain a measure of control over the situation [67]. Thus, it is not surprising that an olfactory cue to masculinity might be more easily imbued upon a person than one to femininity, especially in a set of participants who were largely female; it is possible that the results would differ in a group of male participants, so the generalizability of these findings is restricted.

The relationships observed between the attributes of masculinity, pleasantness, cleanliness, and femininity varied depending upon whether they were measured as pertaining to the hypothetical person or to an odorant. In some cases, similar relationships were observed; for example, the attributes of pleasantness and masculinity were not related to each other, regardless of whether they were rated relative to the odorants or to the hypothetical person. As another example, the positive correlations between the attributes of cleanliness, pleasantness, and femininity previously observed in the literature [56,68] were also observed in the present data for both odorants and the hypothetical person. In other cases, however, the relationships between the attributes differed, depending upon whether they were rated in the context of an odorant alone or within the impression of the hypothetical person. Masculinity and femininity were negatively correlated in the ratings of odorants, suggesting that they perhaps were evaluated as a single dimension; this was not the case when the hypothetical person was considered alone, perhaps reflecting the diversity and complexity of gender experiences. When odors were evaluated alone, higher masculinity was associated with lower ratings of cleanliness, in accordance with the literature [56,68]. However, when considered in the context of the hypothetical person, the opposite relationship was observed: a higher level of masculinity was associated with a higher level of cleanliness. The differences observed in the ratings of these attributes suggest that participants integrated all of the information known about the hypothetical person rather than simply rating the odorant.

There are a number of limitations to the present study. A sample comprised of mainly female university students contributed to difficulties with ecological validity, but it was not the only cause of this limitation; although an attempt was made to induce an impression of a hypothetical person using descriptive words, a silhouette, and an odorant, this is certainly not the same as meeting a new person. Future studies could perhaps use a more realistic methodology with a broader sample to explore these questions in a more ecologically valid way. In addition, all of the odorants in the present study were presented at supra-threshold levels of intensity. Though attempts were made to equate the intensities of the odorants, there was a high level of variability that may have added noise to the present data. Since some previous work, e.g., [69], has shown that context can affect behavior implicitly at low levels of odorant concentration, systematically varying intensity levels and observing the effect on impression formation is another avenue for future investigation.

## 5. Conclusions

People have a tendency toward creating a unified, coherent representation of an individual [70]. Studies have shown that assumptions about the personal traits of others can be influenced by a number of sources of information, including verbal descriptors, e.g., [1], facial images [71,72], and observed behaviors [73]; the current work adds olfactory information to the list. The present finding that odorants, particularly those high in perceived masculinity, can affect first impressions offers the opportunity for raising consciousness in interactions with others. Awareness of the subtle contribution of olfactory information to impression formation allows people both to control how they are perceived in a first impression setting and also to be alert to their perceptions of others. Although odor is certainly not the only information to be considered when forming an impression, it does seem to provide useful evidence about a new person.

## Figures and Tables

**Figure 1 brainsci-11-00955-f001:**
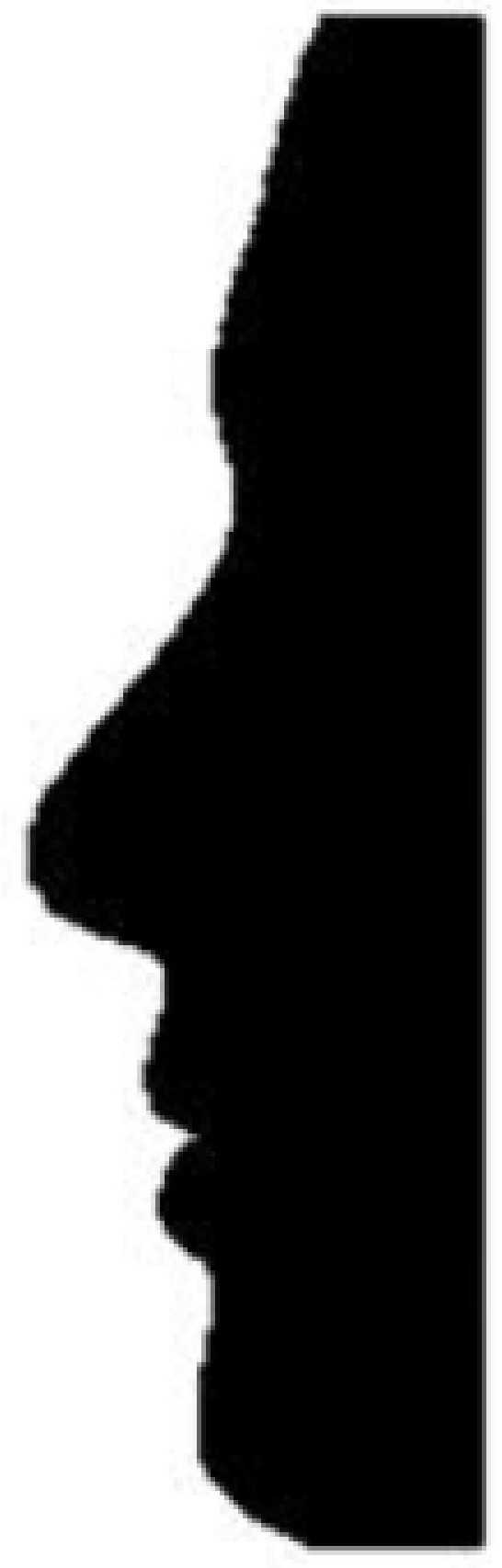
Gender-neutral silhouette [58] presented to all participants.

**Table 1 brainsci-11-00955-t001:** 51 Personality traits used for ratings of a hypothetical person. The traits are listed in the six categories (attributes) determined a priori by the researchers; traits were presented randomly without regard to attribute to each participant during the study.

Clean	Pleasant	Feminine	Unclean	Unpleasant	Masculine
Generous	Wise	Weak	Ungenerous	Shrewd	Strong
Good-natured	Happy	Sensitive	Irritable	Unhappy	Realistic
Pure	Humorous	Delicate	Impure	Humorless	Responsible
Reliable	Sociable	Feminine	Unreliable	Unsociable	Competent
Humane	Popular	Sentimental	Ruthless	Unpopular	Professional
Altruistic	Important	Quiet	Self-centered	Insignificant	Intelligent
Honest	Good looking	Emotional	Dishonest	Unattractive	Logical
	Persistent	Romantic		Unstable	Serious
	Talkative	Sincere		Hard-headed	Career-Oriented
	Imaginative				

**Table 2 brainsci-11-00955-t002:** Mean attribute ratings and mean root mean squared deviations (RMS).

Odor Group	Focus of Rating	Mean Attribute Ratings				
		Pleasantness	Masculinity	Femininity	Cleanliness	RMS (SD)
**Onion**	Hyp. Person	54.98	78.04	41.14	55.44	
	Onion	35.16	63.62	19.26	20.39	29.51 (6.97)
	Lemon	56.30	56.45	42.54	66.67	30.96 (12.94)
	Amber	56.02	28.41	61.05	64.10	38.52 (12.95)
	Lily	70.12	22.69	61.00	83.04	39.03 (10.46)
**Lemon**	Hyp. Person	62.16	73.76	60.23	60.00	
	Onion	42.19	54.17	19.87	21.94	39.82 (11.70)
	Lemon	62.78	44.65	43.94	81.61	37.31 (12.01)
	Amber	57.01	31.52	47.90	72.97	30.63 (11.54)
	Lily	70.48	12.95	73.45	91.13	34.00 (15.42)
**Amber**	Hyp.Person	61.58	78.97	53.11	58.47	
	Onion	39.19	52.43	22.20	21.88	36.22 (12.12)
	Lemon	66.56	51.79	42.54	83.79	39.98 (9.54)
	Amber	57.67	37.83	54.95	67.58	31.15 (10.29)
	Lily	71.62	15.94	64.21	89.51	33.16 (13.19)
**Lily**	Hyp. Person	61.73	80.70	60.75	61.90	
	Onion	37.82	51.15	20.28	13.36	41.22 (11.17)
	Lemon	59.00	51.92	38.50	69.78	37.62 (8.04)
	Amber	61.09	27.32	68.18	79.48	32.83 (9.61)
	Lily	67.13	18.21	60.26	86.69	35.06 (9.73)

**Table 3 brainsci-11-00955-t003:** Mean *Same-vs.-Other* disparities in the principal analysis (composed of estimated ANCOVA means plus odorant intensity effects).

Stimulus Group	ANCOVA *Same-vs.-Other* Disparity Mean	Estimated Intensity Effect	Principal Analysis Mean
Onion	−7.61	0.95	−6.66 **
Lemon	3.39	−0.89	2.50
Amber	−5.76	0.45	−5.31 **
Lily	−1.50	−0.66	−2.16
All Groups	−3.06	0.00	−3.06 **

Note: ** indicates that the mean *same-vs.-other* disparities were significantly less than zero at the 0.01 level (one-tailed).

**Table 4 brainsci-11-00955-t004:** Relationships between attributes both within the context of the hypothetical person and the direct ratings of the averaged odorants.

Attributes Correlated	Person (*r*)	Odor (*r*)
Pleasantness-Masculinity	0.130	−0.073
Pleasantness-Femininity	0.333 *	0.223 **
Pleasantness-Cleanliness	0.437 **	0.392 **
Masculinity-Femininity	−0.131	−0.497 **
Masculinity-Cleanliness	0.275 **	−0.303 **
Femininity-Cleanliness	0.610 **	0.327 **

Note: * indicates that correlation is significant at the 0.05 level (2-tailed). ** indicates that correlation is significant at the 0.01 level (2-tailed).

## Data Availability

The data presented in this study are openly available in Open Science Framework at https://osf.io/mcw4f/?view_only=6c7d5d47752b4ab39ab2cbe4ce9a625d (accessed on 20 July 2021).

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
