# Peer review of "Scent of a Woman—Or Man: Odors Influence Person Knowledge"

_brainsci, 2021, doi:10.3390/brainsci11070955_

Round 1

Reviewer 1 Report

It was a pleasure reading the write up of this interesting and relevant research. I've provided feedback in the hope it may improve the manuscript so I hope my comments are taken in this light.

Abstract

line 15: unclear what "by group" means - a clarification would be helpful

line 16: unclear who "they" and "them" are referring to so I suggest rewording the sentence without using such terms.

I think more and clearer information regarding the methodology is needed to help understand the results. For example, it's unclear how and what odours were presented. What were the odour "qualities"? Giving specific examples of the characteristics of the stimuli would be helpful.

Introduction

Overall, the introduction covers the relevant literature in a clear manner. The hypotheses require a significant improvement. My specific comments are below.

lines 32-33: is this processing specifically related to face processing? If so, this needs to be made clearer

line 40: "The way that people smell" colloquial and unclear if this means BO or olfaction, please clarify.

line 44: more recent and nuanced studies exist than reference 21 exists (e.g., https://doi.org/10.1016/j.physbeh.2018.09.014) so recommend citing

line 76: does "color" refer to clothing color or something else? Please clarify.

lines 79-80: unconvincing and unclear argument on how you get from color to odor as an associative cue. I recommend refer to research on associations formed between colored liquids and odors as a segue to stating colors can imbue perceived characteristics of odors.

lines 101-102: incorrect tense "assess" should be "assessed"

line 113: "same attributes" as what - while I realize the Methods section is coming, I've still not got a handle on what appears to be a key part of the design, presumably, rating personality and odor using the same characteristics. 

lines 114-116: the current H1 should not be an H1 as is dependent on an odor being successfully associated with a person

lines 117-121: the "next" hypothesis, which should be termed "second hypothesis" is long and not clearly articulated. Moreover, it appears and contains H1. These lines need a significant re-working.

lines 121-125: this section is not clearly articulated - please re-word to improve clarity.

Materials and Methods

Overall, this section did not clearly present the procedure or materials used and requires an overhaul. My specific comments are below.

lines 131-132: unclear what "lemon group" and the rest, are. This is the first mention of what I assume is the odor "groups". A clearer unravelling of the design is required.

line 150: unclear what "original E-Prime program" means

line 155: "(a fine fragrance)" not needed

line 160-178: the design of the study is unclear - for example, 

  • it's unclear HOW participants "form[ed] an impression" - just by reading the characteristics?
  • How did the researchers know when a "strong impression "was formed?
  • Did participants receive different lists of characteristics?
  • What is a "context for consideration"? Are "characteristics" and "personality attributes" the same thing?

Unfortunately, the entire Procedure section is unclearly articulated and includes information that shouldn't be in this section (that is, lines 171-175 that presents how/why the characteristics to be rated were selected).

line 182 - I recall there being 5, not 4, olfactory stimuli - I'm assuming no rating of a "blank odor" occurred?

line 184: unclear what "variations of the gLMS" means

Results

line 203: were different scales used to rate different odors?

line 203: Use the same term for olfactory stimuli (here it's "odorants", line 264 it's "scents").

lines 212-216: a better study design would be the use of the same characteristics of odors and persons, with distractors, to disguise purpose of study. This would have removed the need for the numerous transformations.

line 222: stimuli names have changed, for example, lily is now "Lily Fine Fragrance". Please use one label throughout the manuscript

line 234: the label "same" should be "same odor" to more clearly covey what it represents

line 236: Table 2 does not contain labels "same" or "person vs same" as anticipated based on the reading of the paragraph.

line 250: stating what hypothesis is being tested would help the reader understand the purpose of the analyses. With the numerous transformations and the fact that the results are based on disparity ratings, it becomes difficult to map the findings to the hypotheses. It's helpful to have headings that clearly state the purpose of the analyses

line 265: start the paragraph with a heading or a sentence that clearly states the purpose of the analysis and what hypothesis it's addressing

line 280: It is necessary to state the purpose of the intensity ratings analyses as none is presented or discernable. 

line 303: the use of the correlation results as evidence does not need to be limited to "informal" comparisons. The difference between two correlations can be tested using the Fisher r-to-z transformation.

Discussion

Overall, the results to not support the claims made in the discussion. The results should be considered and presented as "preliminary evidence" at best.

lines 351-354: unclear how results demonstrated odors changed perceptions of gender

line 359-360: a silhouette, words and an odor were used which does not con

line 383: "...are not body odors, participants treated the smells as though they were..." no strong evidence for this study's data

No limitations were mentioned but the following should be:

1. ecological validity of the study: making first impressions about a silhouette may not be the same as first impressions of actual people (images or real) and presents an avenue for future research

2. results aren't generalizable to males mentioned as only small percentage of sample were males

3. the number of scale transformations and re-categorizations of the data indicate the design of the study was not robust

References

reference 7 appears to be missing

Author Response

We’re glad that you enjoyed the manuscript.  Thank you very much for taking the time to provide a careful and thorough review.

Abstract

line 15: unclear what "by group" means - a clarification would be helpful

We were referring to the fact that each person only got one odorant, so there were 5 groups.  Given the word limit for the abstract, and the fact that we mention on line 15 that each person only received one of the 5 possible odorants, we removed the statement “by group” to improve clarity.

line 16: unclear who "they" and "them" are referring to so I suggest rewording the sentence without using such terms.

We have attempted to reword the sentence for improved clarity.

I think more and clearer information regarding the methodology is needed to help understand the results. For example, it's unclear how and what odours were presented. What were the odour "qualities"? Giving specific examples of the characteristics of the stimuli would be helpful.

While we agree that this level of detail would be useful to readers, the 200-word limit for this journal makes adding this information to the abstract impossible.

Introduction

Overall, the introduction covers the relevant literature in a clear manner. The hypotheses require a significant improvement. My specific comments are below.

lines 32-33: is this processing specifically related to face processing? If so, this needs to be made clearer

This processing is not specifically related to the processing of faces.  The inclusion of the ESBA and FFA was meant to underscore known brain areas that are responsible for processing physical information about a new person.  Because the specific naming of those areas was confusing, we have removed them.

line 40: "The way that people smell" colloquial and unclear if this means BO or olfaction, please clarify.

This is has been clarified

line 44: more recent and nuanced studies exist than reference 21 exists (e.g., https://doi.org/10.1016/j.physbeh.2018.09.014) so recommend citing

We agree with the reviewer, and have substituted the reference.

line 76: does "color" refer to clothing color or something else? Please clarify.

Generally, in American culture, pink and blue act as distinctive stereotypical cues.  We have attempted to clarify the sentence.

lines 79-80: unconvincing and unclear argument on how you get from color to odor as an associative cue. I recommend refer to research on associations formed between colored liquids and odors as a segue to stating colors can imbue perceived characteristics of odors.

Because this work on triggering gender stereotypes associated with odors is highly relevant to the present work, we have attempted to clarify it, rather than replacing it with associations to colored liquids.

lines 101-102: incorrect tense "assess" should be "assessed"

This has been corrected.

line 113: "same attributes" as what - while I realize the Methods section is coming, I've still not got a handle on what appears to be a key part of the design, presumably, rating personality and odor using the same characteristics. 

We have clarified the section by adding the specific attributes.

lines 114-116: the current H1 should not be an H1 as is dependent on an odor being successfully associated with a person

This hypothesis has been renamed the principal hypothesis.

lines 117-121: the "next" hypothesis, which should be termed "second hypothesis" is long and not clearly articulated. Moreover, it appears and contains H1. These lines need a significant re-working.

We apologize for the lack of clarity and have attempted to improve it. 

lines 121-125: this section is not clearly articulated - please re-word to improve clarity. 

We have shortened the sentence to clarify.

Materials and Methods

Overall, this section did not clearly present the procedure or materials used and requires an overhaul. My specific comments are below.

lines 131-132: unclear what "lemon group" and the rest, are. This is the first mention of what I assume is the odor "groups". A clearer unravelling of the design is required.

We believe this is confusing because it appears before you’ve been able to read about the materials and procedures.  We’ve moved this information to the procedure section to hopefully improve the clarity.

line 150: unclear what "original E-Prime program" means

E-Prime is a software package, and the computerized program for running parts of this study was created using this software.  We have attempted to clarify this in the text.

line 155: "(a fine fragrance)" not needed

The words have been removed.

line 160-178: the design of the study is unclear - for example, 

  • it's unclear HOW participants "form[ed] an impression" - just by reading the characteristics? 

We did not test or regulate the way that participants formed an impression.  We gave them the contextual items listed in the Procedure section (odor, silhouette, and standard list of 6 personality characteristics).  We have attempted to clarify the Method section to reflect this.

  • How did the researchers know when a "strong impression "was formed?

As stated in the Procedure section, the participants indicated when the impression had been formed.  This section has been revised for clarity.

  • Did participants receive different lists of characteristics?

All participants received the same initial list of six personality characteristics initially.  They also rated the same 51 traits for each odorant (although these traits were presented in a different random order for each participant).

  • What is a "context for consideration"?

As stated in the Procedure “… three types of contextual stimuli:  The laminated picture of a gender-neutral silhouette [58] depicting a hypothetical person, the laminated list of the six personality characteristics belonging to the hypothetical person [1], and a plastic container containing an olfactory stimulus (varying by group).”

  • Are "characteristics" and "personality attributes" the same thing?

We have revised the manuscript to be clearer and more consistent with the usage of these words. 

Unfortunately, the entire Procedure section is unclearly articulated and includes information that shouldn't be in this section (that is, lines 171-175 that presents how/why the characteristics to be rated were selected). 

We have clarified the section and feel that the information included in the Method section is all relevant to our study.

line 182 - I recall there being 5, not 4, olfactory stimuli - I'm assuming no rating of a "blank odor" occurred?

You are correct.

line 184: unclear what "variations of the gLMS" means

gLMS refers to the general Labeled Magnitude Scale.  The text has been clarified in this regard.

Results

line 203: were different scales used to rate different odors?

No.  All odors were rated with the same scales, as per the Method section.

line 203: Use the same term for olfactory stimuli (here it's "odorants", line 264 it's "scents").

This has been clarified.

lines 212-216: a better study design would be the use of the same characteristics of odors and persons, with distractors, to disguise purpose of study. This would have removed the need for the numerous transformations.

This is an interesting idea, though we feel that the necessity of transformations would still be present.

line 222: stimuli names have changed, for example, lily is now "Lily Fine Fragrance". Please use one label throughout the manuscript

This has been clarified.

line 234: the label "same" should be "same odor" to more clearly covey what it represents

The correction has been made.

line 236: Table 2 does not contain labels "same" or "person vs same" as anticipated based on the reading of the paragraph.

This section has been clarified.

line 250: stating what hypothesis is being tested would help the reader understand the purpose of the analyses. With the numerous transformations and the fact that the results are based on disparity ratings, it becomes difficult to map the findings to the hypotheses. It's helpful to have headings that clearly state the purpose of the analyses

We have added additional information to that end.

line 265: start the paragraph with a heading or a sentence that clearly states the purpose of the analysis and what hypothesis it's addressing

We hope that we have clarified the analyses.

line 280: It is necessary to state the purpose of the intensity ratings analyses as none is presented or discernable. 

We have rearranged the paragraph to make the purpose more prominent.

line 303: the use of the correlation results as evidence does not need to be limited to "informal" comparisons. The difference between two correlations can be tested using the Fisher r-to-z transformation.

Although the Fisher r-to-z can be used readily in independent samples, given that each of these correlations arise from data given from the same individuals, they can hardly be said to be independent.  A dependent samples version of the test can be calculated, but only if a third correlate that shares one variable in common.  This is not the case in the present data (comparing person and odorant correlations).

Discussion

Overall, the results to not support the claims made in the discussion. The results should be considered and presented as "preliminary evidence" at best.

We respectfully disagree with this statement.

lines 351-354: unclear how results demonstrated odors changed perceptions of gender

See section 3.5.3

line 359-360: a silhouette, words and an odor were used which does not con

We did not make any changes based on this comment, as we did not understand the reviewer’s intent.

line 383: "...are not body odors, participants treated the smells as though they were..." no strong evidence for this study's data

We believe that there is, and have attempted to clarify this point.

No limitations were mentioned but the following should be:

  1. ecological validity of the study: making first impressions about a silhouette may not be the same as first impressions of actual people (images or real) and presents an avenue for future research

A limitations section was added that addresses this very important issue.

  1. results aren't generalizable to males mentioned as only small percentage of sample were males

This issue is addressed in line 433.

  1. the number of scale transformations and re-categorizations of the data indicate the design of the study was not robust

We respectfully disagree with the reviewer on this issue.  The design and scales that were used reflect the best psychophysical tools to ask the questions.  The mathematical transformations that were made were regular and standard; they do not detract from the design.

References

reference 7 appears to be missing

This reference was accidently combined with reference 6.  It has been corrected. 

Reviewer 2 Report

Thank you for the opportunity to review this very well written and interesting manuscript. I have no major criticisms, but offer the following suggestions to improve the clarity:

Aspects of the Methods are a bit unclear. Participants were presented with a list of "six personality characteristics" (listed on p. 3), which appear to have been the same for all participants, but perhaps state that explicitly? Participants were shown a "gender neutral silhouette" - perhaps this should be a figure or at least presented in supplementary materials? Then, participants "were asked to form an impression". Did they just do this in their head? They didn't have to do anything further about that impression at that point, is my understanding.

Once they had their impression they then (immediately?) rated the hypothetical person on 51  "personality attributes". This is where the language gets a bit confusing. The hypothetical person had a set of "characteristics", but the rating was on "attributes". These attributes were then divided into "categories".  And Table 1 refers to "Traits used..." Perhaps the language could be clarified throughout the manuscript, including the Abstract. At the very least, the Table 1 caption could read something like "51 personality attributes presented in the 6 categories determined apriori by the researchers".

Perhaps indicate on line 184 that gLMS = general Labeled Magnitude Scale?

In the Results, the discussion of "person-vs-same" (italicized phrases in section 3.3) is confusing because it doesn't obviously correspond to language in Table 2.  For example, line 243 states "the person-vs-other disparity was labeled for convenience as the same-vs-other disparity". I don't see that labeling anywhere in Table 2. The same confusion continued in the discussion of Table 3. I think I got the point, but it was more challenging to follow than it needs to be. Some attention to using consistent language between the text and tables would be helpful.

Questions:

  • I wondered about the fact that all odors were suprathreshold and whether the impact of lower concentration odors would be the same. Some literature (from EP Köster and others) have shown effects of odors on behavior, but only at very low concentrations. I wonder if this might be worth considering in the context of the current results? Moreover, given that sense of smell was self-reported, which is known not to be a reliable way of testing olfactory function, was there any indication in the data that some participants couldn't smell the odors or at least not as well? On a related note, the mean ratings of intensity seem quite variable given that they were presented at "similar intensities" (line 140). This might be worth a comment.
  • I understand that these data were collected some time ago, but it seems that in current times it might be appropriate to at least address the fact that gender/sex is not binary. There was brief mention of a continuum, but perhaps it is worth adding a statement in the Discussion.
  • The participants were overwhelmingly female. Is it worth a comment that the results could be different for males? This seems important given that masculinity seemed to be a salient characteristic in this set of data. Would this be the same for male participants?  I know the sample size of males is small, but were any differences detected between them and the females?

Very small points:

I noticed a missing author name in the "Citation" on the left panel of the title page.

For the title - is "olfactory information" a necessary phrase? Could it simply be "odors"?

Check formatting of Figure 3.

Author Response

Thank you for your kind words and for taking the time to comment on this manuscript.

 I have no major criticisms, but offer the following suggestions to improve the clarity:

Aspects of the Methods are a bit unclear. Participants were presented with a list of "six personality characteristics" (listed on p. 3), which appear to have been the same for all participants, but perhaps state that explicitly?

We have revised the Procedure section (line 161 to 165) to hopefully clarify the fact that all participants received the same list of personality characteristics.

Participants were shown a "gender neutral silhouette" - perhaps this should be a figure or at least presented in supplementary materials?

Because this silhouette has been previously published (reference 58) by another researcher, I’m not certain with the legalities of including it in the current document; I suspect that permissions may be necessary.  I have made it into a figure that I’ve included, because I agree that it adds to the clarity.  I will ask the Brain Sciences editorial staff to make certain that the inclusion of this figure is possible.

Then, participants "were asked to form an impression". Did they just do this in their head? They didn't have to do anything further about that impression at that point, is my understanding.

Your understanding is correct.

Once they had their impression they then (immediately?) rated the hypothetical person on 51  "personality attributes". This is where the language gets a bit confusing. The hypothetical person had a set of "characteristics", but the rating was on "attributes". These attributes were then divided into "categories".  And Table 1 refers to "Traits used..." Perhaps the language could be clarified throughout the manuscript, including the Abstract. At the very least, the Table 1 caption could read something like "51 personality attributes presented in the 6 categories determined apriori by the researchers".

We have attempted to clarify this language throughout, and are grateful for the suggestion on the Table caption.

Perhaps indicate on line 184 that gLMS = general Labeled Magnitude Scale?

This has been clarified.

In the Results, the discussion of "person-vs-same" (italicized phrases in section 3.3) is confusing because it doesn't obviously correspond to language in Table 2.  For example, line 243 states "the person-vs-other disparity was labeled for convenience as the same-vs-other disparity". I don't see that labeling anywhere in Table 2. The same confusion continued in the discussion of Table 3. I think I got the point, but it was more challenging to follow than it needs to be. Some attention to using consistent language between the text and tables would be helpful.

The reviewer is quite correct, and we apologize for the confusion.  The section has been revised, and the language made more consistent.

Questions:

  • I wondered about the fact that all odors were suprathreshold and whether the impact of lower concentration odors would be the same. Some literature (from EP Köster and others) have shown effects of odors on behavior, but only at very low concentrations. I wonder if this might be worth considering in the context of the current results?

A very interesting idea!  We have now raised it in the discussion section.

  • Moreover, given that sense of smell was self-reported, which is known not to be a reliable way of testing olfactory function, was there any indication in the data that some participants couldn't smell the odors or at least not as well?

Only the people in the “no odor” control group complained of not being able to smell anything – quite justifiably!  So we had no indication that anyone had difficulties with their sense of smell

  • On a related note, the mean ratings of intensity seem quite variable given that they were presented at "similar intensities" (line 140). This might be worth a comment.

We have addressed this in the discussion section.

  • I understand that these data were collected some time ago, but it seems that in current times it might be appropriate to at least address the fact that gender/sex is not binary. There was brief mention of a continuum, but perhaps it is worth adding a statement in the Discussion.

We have added a brief statement to the discussion section.

  • The participants were overwhelmingly female. Is it worth a comment that the results could be different for males? This seems important given that masculinity seemed to be a salient characteristic in this set of data. Would this be the same for male participants?  I know the sample size of males is small, but were any differences detected between them and the females?

We agree that a comment in the discussion section regarding the relevance of the participant group to the outcome is worthwhile and have added a statement to that effect [line 443]. We did not observe any differences between the males and females in the present data, but as you correctly pointed out, men comprise only a small portion of the sample.

Very small points:

I noticed a missing author name in the "Citation" on the left panel of the title page.

This has been corrected; thank you for noticing.

For the title - is "olfactory information" a necessary phrase? Could it simply be "odors"? 

Thank you for the suggestion.  We have modified the title.

Check formatting of Figure 3

We believe that you mean Table 3, and we have attempted to improve the formatting.